# Hair Follicles as a Critical Model for Monitoring the Circadian Clock

**DOI:** 10.3390/ijms24032407

**Published:** 2023-01-26

**Authors:** Li-Ping Liu, Meng-Huan Li, Yun-Wen Zheng

**Affiliations:** 1Department of Dermatology, Affiliated Hospital of Jiangsu University, Zhenjiang 212001, China; 2Institute of Regenerative Medicine, Jiangsu University, Zhenjiang 212001, China; 3Guangdong Provincial Key Laboratory of Large Animal Models for Biomedicine, South China Institute of Large Animal Models for Biomedicine, School of Biotechnology and Health Sciences, Wuyi University, Jiangmen 529020, China; 4Department of Medicinal and Life Sciences, Faculty of Pharmaceutical Sciences, Tokyo University of Science, Noda 278-8510, Japan; 5Department of Regenerative Medicine, Yokohama City University School of Medicine, Yokohama 234-0006, Japan; 6Division of Regenerative Medicine, Center for Stem Cell Biology and Regenerative Medicine, Institute of Medical Science, The University of Tokyo, Tokyo 108-8639, Japan

**Keywords:** circadian rhythm, clock, hair follicle, sleep, aging, chronotherapy

## Abstract

Clock (circadian) genes are heterogeneously expressed in hair follicles (HFs). The genes can be modulated by both the central circadian system and some extrinsic factors, such as light and thyroid hormones. These circadian genes participate in the regulation of several physiological processes of HFs, including hair growth and pigmentation. On the other hand, because peripheral circadian genes are synchronized with the central clock, HFs could provide a noninvasive and practical method for monitoring and evaluating multiple circadian-rhythm-related conditions and disorders among humans, including day and night shifts, sleep–wake disorders, physical activities, energy metabolism, and aging. However, due to the complexity of circadian biology, understanding how intrinsic oscillation operates using peripheral tissues only may be insufficient. Combining HF sampling with multidimensional assays such as detection of body temperature, blood samples, or certain validated questionnaires may be helpful in improving HF applications. Thus, HFs can serve as a critical model for monitoring the circadian clock and can help provide an understanding of the potential mechanisms of circadian-rhythm-related conditions; furthermore, chronotherapy could support personalized treatment scheduling based on the gene expression profile expressed in HFs.

## 1. Introduction

Hair loss affects a large part of the world’s population. Although not life-threatening, hair loss has a significant negative effect on life quality [1] and leads to psychological disturbance, distress, self-consciousness, embarrassment, frustration, and jealousy [2]. Lifestyle, especially sleep quality [3], is thought to be associated with hair loss. Several studies reported that sleep disturbances could cause various immunological reactions in the skin, which may influence the development of hair loss [4,5]. Therefore, sleep seems to affect the growing and shedding of hair follicles (HFs) through certain mechanisms. Sleep–wake cycles are intertwined with the circadian timing system, which modulates behavior and physiology, such as body temperature, metabolism, and hormone secretion in humans; this behavior and physiology fluctuate over approximately 24 h. This periodic fluctuation is a biological phenomenon known as the circadian rhythm [6], and its controlling system is called the circadian clock. In recent years, mounting environmental and lifestyle changes, such as artificial lighting, red-eye flights, and electronic products, have disturbed human sleep, and thus further disorganized the circadian rhythm. The consequences of this disruption are profound and could cause widespread health problems [7], such as cancer, cardiovascular diseases, and metabolic obesity [8]. Although the definite causation between sleep disturbances and hair loss is still uncertain, increasing evidence suggests potential effects of the circadian clock on HF regeneration [9], pigmentation [10], or aging [11]. Furthermore, HFs have a growth cycle that comprises anagen, catagen, and telogen; and this unique periodic feature is reminiscent of intrinsic rhythms. Therefore, a revealment of the internal association between the circadian clock and HFs is essential for a full understanding of the potential mechanisms of hair loss and hair aging. This revealment will help in the development of new therapeutic strategies. In addition, as with most human tissues, HFs express peripheral clock genes; and collecting HF samples can provide a noninvasive way to monitor the central circadian rhythm of the human body [12]. In this study, we focused on the potential effects of the circadian clock on HFs mainly involved in the physiological process. Due to the noninvasive advantages of HF sampling, monitoring human circadian rhythm and an evaluation of associated diseases using HF cells were also performed.

## 2. Expression of Circadian Clock in HFs and Its Effects on the Physiological Process

The circadian clock consists of central and peripheral clocks. The central clock is located in the suprachiasmatic nucleus (SCN) of the hypothalamus, which is the pacing point for circadian rhythms. Light is perceived by the retina and then transmitted to SCN neurons via electrical signals, which release instructions to the organism and affect the peripheral clock system through neurotransmitters, endocrine factors, and bodily fluids [13]. The peripheral clock can be found in almost all peripheral tissues, such as the skin, heart, liver, and kidneys, and is regulated by the central clock to produce synchronized rhythms [14]. At the molecular level, the mechanisms of the circadian rhythms that are generated by the central and peripheral clocks are similar. The circadian clock mechanism is composed of interdependent feedback loops of transcription and translation of specific gene products. The bHLH-PAS transcriptional activators CLOCK and BMAL1 form a heterodimer and activate target genes containing E-boxes in their enhancer regions, including period 1–3 (*PER1/2/3*) and cryptochromes 1 and 2 (*CRY1/2*). PER and CRY proteins are imported into the nucleus and repress the transcription of their own gene loci, whereupon a new circadian cycle can begin. In addition, the CLOCK-BMAL1 heterodimer activates the expression of nuclear receptors ROR and REV-ERBα, which in turn activate and inhibit, respectively, the transcription of *BMAL1* and other target genes containing retinoic-acid-related orphan receptor response elements. The second feedback loop provides additional robustness to the circuitry [15,16] (Figure 1).

### 2.1. Heterogeneous Expression of Clock Genes in HFs and Influence from Extrinsic Factors

HFs move through regular growth cycles throughout their life process which comprises three phases: anagen, catagen, and telogen [17]. Stages of rapid growth and elongation of the hair shaft alternate with periods of quiescence and regression driven by apoptotic signals. The anagen phase is the most active period of HF growth: hair grows rapidly and forms a complete hair shaft. During anagen, transient amplifying cells derived from epithelial hair follicle stem cells (HFSCs) in the bulge proliferate intensively and then differentiate into hair shafts and inner root sheaths [18]. The catagen phase is a short transition stage that occurs at the end of the anagen phase. The catagen phase represents the end of active growth of a hair shaft. During this phase, the low segment of HFs undergoes apoptosis-driven regression. Dermal papilla cells begin to condense and move upward; and when they move to a position underneath the bulge area, HFs enter the telogen phase [19]. In this phase, HFs have the weakest biological activity and the hair shaft falls off. However, the expression and activity of relevant regulators in HFs that control their periodic growth are significantly enhanced in preparation for the beginning of the next anagen. After a period of rest, HFs are stimulated by an external signal. Cells located in the secondary hair germ start to proliferate first, then bulge HFSCs start to proliferate and migrate downward to form transient amplifying cells, and the next anagen stage of HFs begins [20] (Figure 2).

Rhythmic fluctuations have been detected in peripheral tissues including HFs, and circadian-clock-regulation genes exhibit periodic expression in phase with the hair growth cycle. During the critical stage of the initiation of hair growth at telogen and early anagen, enhanced circadian expression of circadian-clock-regulation genes occurs. The secondary hair germ, which contains actively cycling stem and progenitor cells, is where most prominent rhythmic circadian genes are expressed [9]. As anagen progresses, the epithelial matrix and the mesenchymal dermal papilla become critical sites for the circadian activity specific to regenerating anagen HFs [21]. Through microarrays, it has been found that telogen and anagen mice skin present distinct circadian-gene-expression programs [23], indicating a dynamic regulation of the circadian transcriptome within HFs. Furthermore, Janich et al. [24] found that follicular bulge stem cells display inherent circadian heterogeneity to physiological activation, which is modulated by Bmal1. There is a population of “ready” cells that can respond more quickly and efficiently to activating stimuli, while another group is less prone to the stimuli. Deletion of *Bmal1* would result in a progressive accumulation of dormant stem cells, indicating a circadian clock fine-tuning of the temporal behavior of HFSCs.

Oscillating molecular clock activity in peripheral tissues is synchronized with the central clock located in the SCN. Thus, factors that can influence the central clock may also affect peripheral tissues, including HFs, directly or indirectly. For example, external light regulates daily rhythmic tissue activities by entraining the central circadian clock through the eyes [25,26]. This process is performed by a group of photoreceptors located in the intrinsically photosensitive retinal ganglion cells (ipRGCs) [27,28]. It has been found that external light stimulation of the eyes of C57BL/6 mice results in rapid activation of HFSCs and prominent hair regeneration, which is interpreted by an autonomic nervous system circuit of ipRGCs [29]. Ex vivo experiments also show the positive impact of blue light on hair growth, and CRY1 might mediate these effects [30]. In addition to light, thyroid hormones are recognized as key regulators of circadian rhythmicity [31,32], which is crucial for seasonal rhythms and the timing of the mating season in mammals [33]. HFs are direct targets of thyroid hormones. Thyroxine T4 treatment can prolong the anagen of cultured human HFs in vitro, and both T3 and T4 significantly stimulate intrafollicular melanin synthesis [34]. Moreover, thyroid hormones can directly modulate peripheral clock activity in human HFs in the absence of central clock inputs. It was found that both transcript and protein levels of core clock genes, *BMAL1* and *PER1*, were significantly decreased in cultured human HFs after 24 h of T4 treatment, although their rhythmicity was still maintained [35]. By contrast, a 6-day treatment resulted in the upregulation of transcript and/or protein levels of BMAL1, PER1, CLOCK, CRY1, and CRY2. 

Thus, circadian clock genes in HFs show heterogeneous expressions depending on hair cycle and cell types (Figure 2). Extrinsic factors such as light and thyroid hormones also have influence on expression levels of these clock genes. 

### 2.2. Regulatory Role of Clock Genes in the Hair Cycle

Both the regenerative cycling of HFs and the circadian clock show defined periodic features, and a connection should exist between these two timing systems. To explore the role of clock genes in the regulation of the hair cycle, different gene mutant mice models were established. In global *Bmal1−/−* mice, a clear delay at the first anagen phase of HFs was found [9], suggesting an important role of clock genes in the synchronized HF cycling of growth. However, K14Cre mediated keratinocyte *Bmal1* selective deletion is insufficient to reproduce this anagen delay [22]. Thus, BMAL1 intrinsic to keratinocytes may be dispensable in normal anagen initiation of HFs. Recently, Ray et al. [36] found that rhythmic changes in the transcriptome, proteome, and phosphorylated proteome were still present in *Bmal1* knockout mice, indicating that deletion of the key clock gene *Bmal1* does not affect the circadian oscillations of all rhythm molecules. This periodic oscillation may be caused by transcriptional regulation mediated by the erythroblast transformation specific transcription factor family. Therefore, there may be another set of rhythmic oscillation “pacemakers” in HFs, which probably explain the results found in *Bmal1fl/fl*;*K14Cre* mice. In addition to BMAL1, PER1 shows effects on the hair cycle; and the knockdown of *PER1* led to significant anagen prolongation of cultured human HFs [22].

Since the circadian clock is implicated in hair cycle control, what is the potential mechanism of this control? To address this question, the cell cycle was examined in global *Bmal1−/−* mice HFs; and an absence of mitotic cells was found within the delayed HFs at anagen, which is likely due to a block at the G1 phase of the cell cycle in the secondary hair germ cells [9]. By contrast, constant and elevated cell proliferation were detected in the interfollicular epidermis and upper HFs in *Bmal1fl/fl;K14Cre* mice [23], suggesting that BMAL1 intrinsic to keratinocytes is required for time-of-the-day-dependent proliferation during homeostatic cell division. Using an inducible keratinocyte-specific Bmal1 deletion mouse model, it was found that the clock coordinates cell cycle progression by synchronizing G2/M checkpoint, which makes hairs grow faster in the morning than they do in the evening [21]. Since DNA is more susceptible to damage during cellular mitosis, circadian mitotic gating may provide an important protective mechanism in highly proliferative HFs. Thus, circadian clock genes could control HF cycling by regulating the cell cycle and proliferation.

In addition, it was found that the knockdown of *BMAL1* in HF dermal papilla cells resulted in significant decreases in mRNA expression of the hair-growth-related genes *WNT10B*, *LEF1*, *STAT3*, and *BMP4* [37]. Interestingly, the level of the androgen receptor also decreased. Due to the important role of the androgen receptor in the pathogenesis of androgenetic alopecia, these results may lead to the introduction of new therapies, such as chronotherapy, in the modulation of the circadian gene expression in alopecia treatments.

### 2.3. Involvement of the Circadian Clock in HF Aging

Aging is a complex process that is influenced by both genetic and environmental factors. Growing evidence indicates an association between the circadian clock and the aging process. For example, *Bmal1−/−* mice showed a significantly shorter lifespan compared to wild-type mice and exhibited a range of premature aging phenotypes, including hair regeneration defects, age-related lens and cornea defects, and reduced subcutaneous fat [11]. In old *Bmal1* mutant mice, hair regeneration after shaving was significantly delayed compared with regeneration in wild-type mice. However, this phenomenon was not found in young knockout mice, indicating that it was an age-related defect [11]. In another aging mouse model, which was established by a localized irradiation regimen and closely mimicked many aspects of physiological skin aging, several features of HF aging, such as striking baldness, HF miniaturization, and hair graying, were observed [38]. These changes may be associated with CLOCK because inhibition of the miR-31-CLOCK-MAPK/ERK pathway could ameliorate premature aging effects [38], indicating the involvement of circadian genes in HF aging.

Normal homeostatic functions of adult stem cells have rhythmic oscillations, which could become arrhythmic during aging. *CLOCK* gene levels of primary human mesenchymal stem cells (hMSCs) derived from healthy donors at different ages decrease with both passage and age. CLOCK deficiency accelerates hMSC senescence, whereas its overexpression rejuvenates physiologically and pathologically aged hMSCs [39]. Thus, CLOCK may play a potentially protective role during hMSC aging. What is the situation with HF aging? HFSCs in the bulge provide a source of hair regeneration, and these stem cells have been found to leave their niche during HF aging and terminally differentiate into epidermal cells [40,41]. This process is called transepidermal differentiation; it could lead to HF miniaturization and, ultimately, baldness. However, the association between circadian disruption and the aging of HFSCs is uncertain. In the premature mice model mentioned above that was established by localized irradiation, it was shown that miR-31 upregulation could drive HFSC transepidermal differentiation, and this process was associated with the circadian gene *Clock* [38]. In addition, using a circadian clock reporter mouse model, it was shown that Bmal1 modulated the expression of regulatory genes of the epidermal stem cell in an oscillatory manner and that arrhythmia of these stem cells led to premature aging [24]. 

In addition, telomere length and telomerase activity are potential markers of biologic skin aging [42]. Telomere shortening impairs the ability of mobilization and proliferation of epidermal stem cells in HFs and leads to an aging phenotype of hair graying and hair loss [43], while overexpression of telomerase reverse transcriptase (*TERT*) or telomerase reintroduction can correct these defects, facilitate robust hair growth, and prevent degenerative pathologies [44,45,46]. Indeed, telomerase activity and telomere length dynamics are under regulation of the circadian clock, and telomerase activity exhibits endogenous circadian rhythmicity. It was found that the mRNA level of *TERT* oscillates with circadian rhythms that are under the control of CLOCK–BMAL1 heterodimers, and CLOCK deficiency in mice causes loss of rhythmic telomerase activities, TERT mRNA oscillation, and shortened telomere length [47]. 

At the molecular level, aging is highly associated with increased levels of reactive oxygen species (ROS) [48]. Therefore, circadian proteins that participate in DNA repair and the accumulation of cellular ROS also have effects on aging [49]. It was found that Bmal1-deficient mice accumulate significantly more ROS than do the control group, which can potentially explain the phenotype of progeria [11]. However, the involved process has not been evaluated in HF aging. 

### 2.4. Association of the Circadian Clock with Hair Pigmentation

The skin has a complex circadian organization, and several types of cells in the skin are likely to act in coordination to drive rhythmic functions within the skin. Functional oscillators and significant amplitudes of clock genes, including *PER2*, *BMAL1*, and *CRY1*, were detected in human keratinocytes, fibroblasts, and melanocytes [50]. Furthermore, melanocytes expressed a photosensitive system mediated by neuropsin [51], which can respond to light stimuli and stimulate the expression of clock genes [52]. Therefore, melanocytes may play a unique role in the regulation of peripheral circadian clock photoentrainment, and not only act in the pigment-associated process. Unlike the state of interfollicular melanocytes, the state of melanocytes in HFs is tightly coupled to the hair growth cycle. Melanocytes become melanogenically active during anagen. When HFs enter catagen, melanogenesis is switched off, and it reappears during the next anagen. Given that circadian genes regulate hair cycle and hair growth, is it possible that they participate in the regulation of hair melanocytes and influence HF pigmentation? It was found that silencing *BMAL1* or *PER1* not only stimulates the melanin content of human HFs but also increases tyrosinase expression and activity [10]. In addition, *TYRP1* and *TYRP2* mRNA levels, gp100 protein expression, melanocyte dendricity, and the number of gp100+ HF melanocytes are increased, indicating a novel role for peripheral circadian clock processes in the regulation of HF pigmentation [53]. 

Melatonin, which is released by the SCN in humans and secreted at night under a robust circadian rhythm, induces night-state physiological functions. There is an increasing focus on the relationship between melatonin and circadian-rhythm-associated disorders. Several placebo-controlled, randomized, clinical data also demonstrated that melatonin treatment has meaningful effects on health problems associated with a disturbed circadian rhythm [54,55]. Interestingly, melatonin and its metabolites inhibit melanogenesis by reducing tyrosinase activity and melanocyte proliferation by stimulating melatonin membrane receptors. These putative effects of melatonin may occur via melatonin-regulated changes in peripheral clock genes such as *BMAL1* and *PER1* [56]. However, the current evidence about the potential role of the circadian clock in melanogenesis and HF pigmentation is still limited.

## 3. Monitoring of Circadian-Rhythm-Related Conditions and Disorders Using HFs 

An increasing body of evidence indicates a correlation between circadian dysfunction and the incidence of sleep disorders, metabolic syndromes, cardiovascular diseases, and cancer [57]. Thus, monitoring the human circadian clock could be an effective approach to study rhythm-related diseases in the clinical setting. Dim-light melatonin onset assay is regarded as the most reliable tool and the gold standard for assessing the human circadian phase [58]; however, it is difficult to standardize and perform this assay at scale because it requires a subject to sit in a dim room for repeated saliva sample collection. Since circadian clock genes are expressed in almost all peripheral tissues while numerous cell-autonomous peripheral clocks are synchronized and governed by SCN [59], it is possible to characterize the human circadian clock by determining gene expression in peripheral tissues. Oral mucosa [60,61] or white blood cell [62,63] samples have been used for this purpose; however, disadvantages still exist. For example, it is difficult to obtain enough living cells and high-quality RNA from the oral mucosa, while physical stimuli and time delays due to the processing of blood cell separation may affect the expression levels of clock genes [64]. Therefore, a noninvasive and reliable method is urgently needed.

HFs are widely used for biological research in areas ranging from molecular biology and stem cell biology to systems biology [65]. As mentioned above, circadian-rhythm-related genes are expressed in HFs and show rhythmic fluctuations that are synchronized with the central clock located in the SCN. Thus, HFs can serve as a great tool for chronobiological studies. A noninvasive method developed by Akashi et al., using cells that remained attached to plucked hairs, was applied to sample collection and circadian-related gene detection [12]. The results can reflect individual behavioral rhythms, which coincide with those detected by the melatonin and cortisol rhythms, indicating the method’s reliability and accuracy. In addition, ex vivo cultures of whole-hair root tissue were established [66,67]; these cultures can provide a convenient method for estimating in vivo circadian characteristics. We summarized an evaluation of the circadian rhythm using the human HF model for various conditions and disorders (Table 1).

### 3.1. Day and Night Shifts

Irregular work schedules have become increasingly common in most industrialized societies. Epidemiological surveys have found that female workers who frequently have rotating work schedules or work at night are more likely to develop endometrial and breast cancers [80,81]. However, the potential pathogenesis remains unexplored. When people who work in shifts cannot adjust their sleep–wake cycle to the light/dark cycle, desynchronization of the circadian rhythm may occur. Multiple studies have showed that markers, such as melatonin, cortisol, and body temperature, of the central circadian pacemaker are reduced in amplitude or distorted when people work atypically—especially during night shifts [70,82,83,84,85]. 

Using keratinocytes and dermal papilla stem cells isolated from HFs of female shift workers, it was found that chronic circadian dysregulation due to day and night shifts affects the expression of *PER1* and *BMAL1* [68]. In addition, there was a loss of clonogenic potential in keratinocytes derived from both the epidermis and HFs in the shift group, suggesting that long-term deregulated circadian rhythm can affect the regenerative properties of human skin and hair precursor cells. In another study using HFs directly for evaluation, a serious time lag between circadian gene expression rhythms and lifestyle in workers with different rotating shift types was also found [12]. To further assess the effects of different shift types on the circadian clock, men with different work time arrangements were recruited; an assay of HF samples indicated that the proportion of workers showing *PER3* and *NR1D2* expression conforming to a 24 h period cosine curve was significantly lower in the continuous night group than it was in the overnight group [69]. In addition, clock gene *PER3* expression patterns of most continuous night shift workers deviated strongly from the 24 h period cosine pattern. It was suggested that the effects of shift work on clock gene expression vary depending on the shift system. Similarly, it was found that after a day off, shift nurses exhibited lower circadian variations than did daytime nurses, using evaluation of peripheral skin temperature and cortisol levels. In addition, maximum values of *PER2* detected using HF cells were significantly different between the two groups [70]. 

Therefore, the detection of individual circadian clock expression patterns by HF cells could be helpful in assessing the status of shift work, which may help in further optimization of shift work schedules.

### 3.2. Physical Activity and Human Behavior

#### 3.2.1. Sleep–Wake Disorders

Sleep problems caused by circadian rhythm disruption are known as circadian rhythm sleep–wake disorders (CRSWDs). They are common in the general population although they are often misdiagnosed. A recent nationwide epidemiological survey conducted in Cyprus revealed that 12.8% of the enrolled participants met the criteria for CRSWD [86]. There are several subtypes of CRSWDs, and patients are typically characterized by a desynchrony between habitual sleep timing and social time schedules [87,88]. Delayed sleep–wake phase disorder (DSWPD) is the most common type which is featured by delayed sleep onset and offset compared to sleep times of healthy subjects. By detecting clock gene expression rhythms using beard follicle cells, it was found that peak time in the DSWPD group was delayed and negatively correlated with sleep quality [71]. In addition, amplitudes of the clock gene, especially *PER3*, positively respond to favorable mental and physical conditions as well as high-quality sleep, indicating that the clock gene may help us understand sleep rhythm disturbances. In addition to DSWPD, there is a group of patients who have a habitual sleep timing that is synchronized with social time schedules but not with circadian-driven sleepiness. This sleep condition is described as latent CRSWD (LCRSWD) [72]. Because patients with LCRSWD have an apparently normal habitual sleep schedule, conventional diagnostic methods for typical CRSWDs are insufficient for detecting it. Using HF samples collected at approximately 8 h intervals, Akashi et al. [72] established a set of criteria to detect the circadian rhythms and sleep parameters of subjects with potential LCRSWD. Through follow-up studies, they assessed the reliability of these criteria and confirmed the reproducibility of candidate screening. Further sleep improvement was also achieved by circadian amelioration, supporting a belief in the reliability of this method. 

#### 3.2.2. Daily Activities

Circadian clock genes regulate daily physiological and behavioral rhythms, such as locomotion, feeding times, sleep–wake cycles, body temperature, metabolism, and hormone-secretion-related activities in humans [89,90]. Conversely, peripheral circadian rhythms can be regulated by scheduled feeding and exercise [91]. Takahashi et al. [73] conducted a cross-sectional study involving 20 human adult males and detected their circadian clock using HF cells. It was found that the amplitude of *PER3* expression was positively correlated with moderate and vigorous physical activity and peak oxygen uptake, suggesting that increasing daily physical activity may affect the in vivo circadian clock system driven by PER3. Similarly, to observe the effect of intensive exercise on clock gene expression, facial hair samples of participants were collected to detect the rhythm of clock gene expression [74]. It was found that, compared with the nonexercise period, the subjects’ circadian clock gene expression was delayed by 2 to 4 h during habitual nighttime training (from 20:00 to 22:00). However, nighttime training did not affect the maximum level and circadian amplitude of clock gene expression. 

#### 3.2.3. Chronotype and Decision-Making

Variation in the timing of endogenous rhythms between individuals results in diverse chronotypes, which can be classified into three categories, namely, morningness, intermediate, and eveningness. There is a robust association between chronotype and social jetlag [92], which is the discrepancy between internal circadian clocks and social clocks. Using HF cells collected from 24 young men in both Friday and Monday trials, it was also found that there was a significant main effect of time on the expression of *NR1D1*, *NR1D2*, and *PER3* in the morningness group but not in the eveningness group, indicating chronotype and social jetlag influence circadian clock gene expression [75]. 

Chronotypes are not only clearly associated with sleep, diet, and physical activity [92,93] but also lead to distinct behavioral responses to stimuli, depending on the time of day [94]. For example, morning chronotypes perform better at cognitive tasks in the morning while better attention and alertness during evening hours have been found in evening chronotypes [95]. Peripheral tissues such as buccal mucosa cells have been successfully utilized in measuring oscillations in circadian gene expression to understand how differences in the molecular clock between individuals are associated with chronotype [96]; however, little is known about how differences in the timing or phase of oscillations of the molecular clock affect human behavior. Ingram et al. [76] set up two decision-making tasks that needed to be completed by participants either in the morning or in the evening. Meanwhile, the chronotypes of the participants were measured through both self-report morningness–eveningness questionnaire (MEQ) and analyses of circadian-clock-regulated genes using HF samples. It was found that, compared to MEQ, HF detection-based chronotypes showed more pronounced effects of chronotype and/or time of the day on decision-making. Therefore, it is feasible to evaluate interindividual differences in human behavior using HF tissue.

#### 3.2.4. Energy Metabolism

The circadian clock plays a critical role in many biological processes, including energy metabolism [97]. Remarkable differences have been found between morning and evening metabolic states, including insulin secretion and sensitivity, and these differences are also associated with clock gene expressions in HFs [98]. In addition, timed food intake affects the human circadian clock [99], and it has been found that continuous total parenteral nutrition (TPN) around the clock may potentially perturb peripheral circadian rhythms. Detection of clock gene expression using HF cells indicated that a portion of aged subjects receiving continuous TPN had abnormal circadian rhythms in peripheral clocks. In contrast, there were no apparent defects among non-TPN subjects [77]. Therefore, a deep knowledge of these associations might be useful for ameliorating metabolic disorders and optimizing TPN administration.

Time of feeding is an important regulator of circadian rhythm [100,101]. Methods, including detection of plasma melatonin and cortisol level and clock gene expression in white adipose tissue or blood, have been used to evaluate the effect of meal time on circadian rhythm [99,102]. However, HFs have not been used in this field. 

### 3.3. Age-Related Diseases

Numerous animal studies have indicated that aging leads to abnormal circadian physiology and behavior which is due to reduced neuronal excitability and fragile intracellular networks in SCN [103]. In humans, age-related abnormalities in circadian rhythms are similar to those in mice [104]. In a recent cohort study that recruited 2930 men aged over 65 years, an association was identified between reduced circadian rhythmicity and an increased risk of Parkinson’s disease (PD)—a typical age-related neurodegenerative disorder [105]. Evaluation of circadian disruption was performed using wrist actigraphy in this study. However, when circadian rhythms were evaluated using plucked hairs, it was found that peripheral clocks of elderly subjects with severe dementia and abnormal circadian behavior oscillated similarly to those of healthy or young subjects [66,78]. Therefore, the circadian pathway from external inputs to the peripheral clock may function normally even in old adults with severe dementia. In PD patients, sleep disturbances are highly prevalent and are regarded as circadian dysfunction because dopaminergic therapy (DT)-receiving PD patients show a circadian phase advance. Recently, the efficacy of bright light therapy in treating sleep problems among DT-receiving PD patients was confirmed in several clinical trials [106,107]. Furthermore, a circadian phase shift of clock gene expression was detected using HFs in most PD patients, providing additional evidence of the correlation between circadian modulation and sleep improvement [79].

### 3.4. Optimization of Evaluation Using HF Cells

As mentioned above, HF sampling offers a noninvasive and practical method for detecting intrinsic human circadian rhythm in the evaluation of various conditions and disorders. However, to guarantee reliable results, multiple rounds of sample collection, for example, every 3–4 h, are necessary. Unfortunately, this is unfeasible and inconvenient in a clinical setting. Thus, sampling at few time points is expected. Akashi et al. [12] attempted to establish a three-point phase prediction. They found that the maximum error in relation to calculated nine-point curves was approximately 2.5 h, and the errors were within 1.5 h in most cases. Furthermore, the tolerance against data fluctuations was extremely high. Furthermore, one-point phase prediction was proposed as a possibility for human data with the standard-curve method, using additional genes whose rhythms of expression can be reliably detected. Moreover, Lee et al. [108] developed an endogenous circadian rhythm estimation model, which combined the relative expression patterns of various circadian clock genes and a machine learning process with a validation algorithm. The model helps to estimate an individual’s endogenous circadian time using only a single time point to sample HF cells. Recently, a user-friendly app, SkinPhaser, was developed to predict the molecular clock phase using the expression values of 12 biomarker genes from one epidermis sample for each tested subject [109]. Given the similarity of cell types in the epidermis and HFs, a software or an app that uses only one HF sample but has a high prediction accuracy and low invasion is expected soon.

## 4. Limitations and Challenges

Current studies have revealed the close association between the circadian clock and HFs, but the complex molecular mechanisms underlying their interactions remain largely unclear. In addition, a growing body of evidence shows that circadian rhythm is associated with sex differences [110,111,112,113]. Revealment of the differences between females and males in circadian timing systems can help give an understanding of some conditions, such as habitual sleep duration and insomnia prevalence, and improve treatment strategies. However, sex difference was not compared in current studies using HF sampling. Some studies only included female participants [68,70], while others only evaluated males (Table 1) [69,71,73,74,75]. Although both genders were included in the remaining studies [72,76,77,78,79], gender is not considered to be an influencing factor. Modification of recruitment strategy to maintain a balance between sexes and carrying out statistical analyses to evaluate the effect of sex are necessary in the future [114]. 

Moreover, due to the complexity of circadian biology, using peripheral human tissues independent of the SCN is insufficiently understood regarding intrinsic oscillatory behaviors. Thus, it remains a major challenge for translational chronobiological research to determine the clinical relevance of data based on HF samples. Combined with HF cells, multidimensional assays, such as detection of body temperature, collection of blood samples, or use of certain validated questionnaires, may be helpful in improving the application of HF data in clinical medicine. Furthermore, optimization of algorithms is necessary and could provide an accurate and precise prediction model for the general population. 

## 5. Conclusions and Perspectives

As mini-organs in the peripheral tissue, HFs express a set of heterogeneous clock genes that can be modulated by both the central circadian system and other extrinsic factors, such as light and thyroid hormones. Circadian genes participate in the regulation of several physical activities of HFs, including the hair cycle, hair aging, and pigment production. On the other hand, because circadian genes expressed in peripheral tissues are synchronized with the central clock located in the SCN, HFs provide a noninvasive and simple method for evaluating individual circadian rhythm traits. HFs can not only help in gaining an understanding of the human biological clock system but also in identifying people at increased risk of circadian-rhythm-related diseases [108]. In addition, because telomere and telomerase activity serve as a link between circadian systems and aging, HFs can provide a practical tool for research into telomere- and aging-related diseases [43]. 

Although the relationship between hair loss and the circadian clock is still uncertain, understanding the traits of circadian rhythm in HFs will help in the development of novel treatments for androgenetic alopecia, telogen effluvium, and hair graying. Several epidemiological studies found that patients with sleep disorders showed an increased risk of developing alopecia areata [4,5]. Regarding pathogenesis, alopecia areata is a complicated autoimmune and inflammatory disease involving Type 1, Type 2, and Th17 axis [115]. The inflammatory cells and the cytokines produced are also under circadian control [116,117]. Therefore, circadian rhythm may play a role in the development of alopecia areata. This role needs to be investigated in a future study. 

Personalized medicine aims to provide patient-tailored therapeutics for improved treatment outcomes because of the great variability of patient responses to therapy. This large variation is partly due to the time-dependent participation of circadian-regulated genes in drug transport and metabolism. In addition, several drugs target circadian-regulated genes [118]. Therefore, chronotherapy, which aims to treat illnesses according to endogenous biological rhythms, has been proposed. Improved patient outcomes with circadian-based treatments have been demonstrated in several clinical trials, especially for cancer and inflammatory diseases [119,120]. Currently, several platforms allow monitoring of circadian biomarkers in individual patients through wearable technologies (rest–activity and body temperature), blood or salivary samples (melatonin and cortisol), and daily questionnaires (food intake) [121]. Therefore, the establishment of an optimized evaluation model using HF sampling would also support personalized treatment scheduling by identifying patients most likely to benefit from circadian-rhythm-based therapies and predicting optimal drug timing based on the patients’ gene expression profile. 

With the increase in industrialization and the level of stress in the world, the accelerated pace of modern lifestyles leads to constant circadian disturbances of the endogenous homeostasis of an organism. Phototherapy or exercise that provides circadian modulation could become a nonpharmacological intervention strategy to counteract the adverse effects of shift work or jet lags. This strategy may offer new possibilities for clinical treatment—not only for hair loss and hair graying, but also for other circadian-rhythm-related disorders. Recently, the therapeutic potential of topical T4 in dermatological therapy, especially in wound healing and hair growth, was proposed [122] due to the role of thyroid hormone receptor-mediated signaling in skin physiology and pathology. Given the association between the circadian rhythm and HFs and the direct modulation of clock genes in HFs by T4, further exploration of thyroid hormone may provide a novel low-cost and effective therapy. 

## Figures and Tables

**Figure 1 ijms-24-02407-f001:**
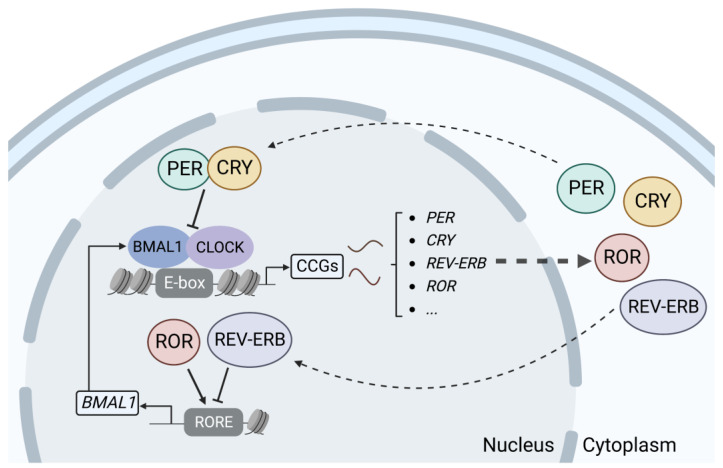
Feedback loops of transcription and translation in the circadian clock system. (Created with BioRender.com).

**Figure 2 ijms-24-02407-f002:**
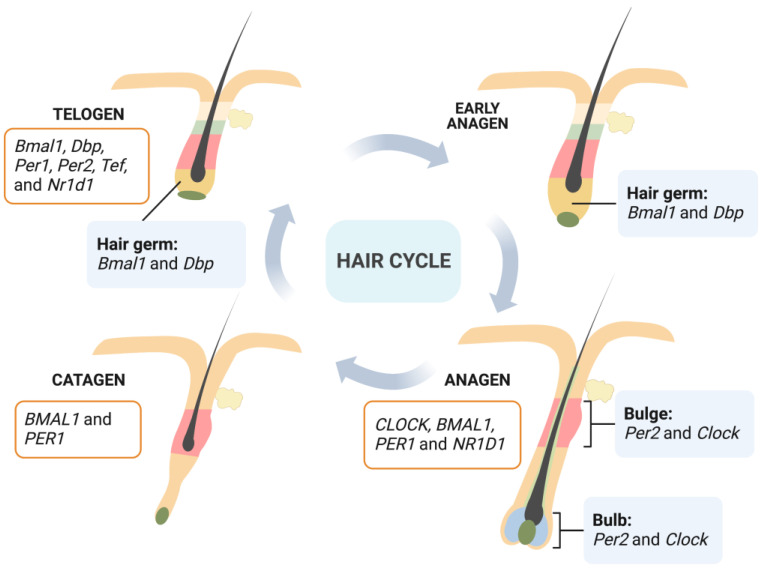
Heterogeneous expressions of clock genes in HFs depending on cell types (bulge and bulb at anagen [21]; hair germ at early anagen and telogen [9]) and hair cycle (anagen and catagen [22]; telogen [9]). Human gene names are italicized, with all letters in uppercase; and mouse gene names are italicized, with only the first letter in uppercase and the remaining letters in lowercase. (Created with BioRender.com).

**Table 1 ijms-24-02407-t001:** Evaluation of the circadian rhythm in various conditions and disorders using human hair follicles.

Conditions/Disorders	Participants (Number)	Tissue/Cells & Collection Method	Observations		Reference
			Circadian Markers	Results	
Day and night shifts	Female diurnal workers (10) and shift workers (10)	Keratinocytes and dermal papilla stem cells isolated from HFs	PER1 BMAL1	Protein expression increased in the deregulated group	[68]
	Rotating shift workers (6)	Scalp HFs; 3 h intervals	*PER2* *PER3* *NR1D1* *NR1D2*	Phase: ~2 h delay	[12]
	Male daytime workers (11), nurses and doctors (10), and factory workers (11)	Beard HFs; 4 h intervals	*PER3*	Mesor: decrease after consecutive night shifts Amplitude: fewer workers with significant 24 h rhythms after night shifts Phase: n.s.	[69]
			*NR1D1*	Mesor: n.s. Amplitude: n.s. Phase: n.s.	
			*NR1D2*	Mesor: decrease after one night shift Amplitude: fewer workers with significant 24 h rhythms after night shifts Phase: n.s.	
	Female shift (23) and daytime nurses (25)	Pubic HFs; Five time points	*PER2*	Shift nurses exhibited lower circadian variations than did daytime nurses	[70]
Sleep disorders	Healthy male participants (16) and male patients with the delayed sleep–wake phase disorder (18)	Beard HFs; 4 h intervals	*PER3*	Amplitude: n.s. Phase: delayed in the group with delayed sleep-wake phase disorder	[71]
			*NR1D1*	Amplitude: n.s. Phase: n.s.	
			*NR1D2*	Amplitude: n.s. Phase: n.s.	
	Latent circadian rhythm sleep–wake disorders (24)	Scalp HFs; 8 h intervals	*PER3*	Peak time of *PER3* expression and GUw are positively correlated	[72]
Physical activity	Adult males (20)	Facial HFs; 4 h intervals	*PER3*	Amplitude: higher in the active group	[73]
			*NR1D1*	Amplitude: n.s.	
			*NR1D2*	Amplitude: n.s.	
	Healthy male (1)	Facial HFs; 4 h intervals	*PER3* *NR1D1* *NR1D2*	Mesor: n.s. Amplitude: n.s. Phase: 2–4 h delay during habitual nighttime training	[74]
Chronotype	Young men (24)	Facial HFs; 4 h intervals	*PER3* *NR1D1* *NR1D2*	Peak time of *PER3* expression was delayed in the eveningness group; change in the peak time of *NR1D1* and *PER3* expression from the Friday to Monday trial was high in the eveningness group	[75]
Decision-making	Extreme morning (16) and evening chronotype (8)	Plucked hairs; 8 h intervals	*PER3* *NR1D2*	3 h phase differences between morning and evening chronotypes	[76]
Energy metabolism	Total parenteral nutrition (TPN) (78) and non-TPN aged subjects (6)	Scalp HFs; 6 h intervals	*PER3*	Peak time of *PER3* relative to wake-up time was delayed in TPN patients	[77]
Age-related diseases	Nondementia elderly subjects (10) and old–old dementia patients (5)	Scalp HFs; 6 h intervals	*PER3* *NR1D1* *NR1D2*	Peripheral circadian phases in dementia subjects were within normal range	[78]
	Parkinson’s disease patients (17)	Scalp HFs; 6 h intervals	*PER3* *NR1D1* *NR1D2*	Phase: delayed after bright light therapy	[79]

Abbreviations: HFs—hair follicles; n.s.—nonsignificant; GUw—local time of getting out of bed on working or school days. Gene names are italicized and protein names are not.

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
