# Peer review of "Hair Follicles as a Critical Model for Monitoring the Circadian Clock"

_ijms, 2023, doi:10.3390/ijms24032407_

Round 1
Reviewer 1 Report
Ms-ijms-2057736 can be accepted with minor revisions.
The data reported by the authors in the review are exhaustive and supported by a substantial bibliography, making a significant contribution to subsequent studies. The work is also well structured in its parts but there is no important aspect of the modulation of circadian genes, that is the dynamics of telomerase and telomeres on stem cells such as those of hair follicles. I invite the authors to add this aspect at least in the final discussion.
The text needs a careful rereading due to the presence of some typos
Reviewer 2 Report
The review paper titled “Hair follicles as a critical model for monitoring the circadian clock,” by Liu et al., summarizes the published literature regarding circadian clock expression and regulation of processes in hair follicle cells. The authors state what is known and still unknown in this field, and what needs to be done in order for hair follicles to be useful in monitoring the circadian clock and establishing an individuals chronotype through non-invasive methods.
- It would be helpful for readers not familiar with the growth process of hair follicles to be provided information on the hair follicle cycle and what the three phases are. The authors state the 3 phases in the introduction, but do not explain what they are or their importance/function/regulation.
- It might be helpful for readers to include an image of the transcription/translation feedback loop that regulates the circadian clock with a bit more explanation that what is described in link 77-79. Additionally, the entire mechanism is not negative. Some parts of the clock mechanism are positive regulation mechanisms.
- In lines 81 and 84, do the authors mean clock-regulated genes (as in circadian clock)? The word CLOCK is capitalized, meaning they are referring to the protein, but it would be more appropriate to say circadian clock-regulation genes.
- Can the authors clarify what they mean by “clockhigh” and “clocklow” cells in lines 92-94?
- For Figure 1: Can the authors clarify if this figure is showing peak expression of these clock genes in hair follicles or if this is showing what stages certain clock proteins seem to be involved it? If it is the latter, it would be good to include the citations in the image for easy reference.
- The authors discussion the relationship with hair loss and circadian clock disturbances. Since the authors propose using hair follicles as a non-invasive way to monitor circadian rhythms, can the authors discuss how such a method could be applied to individuals with extreme hair loss or disorders such as Alopecia areata. Additionally, has there been any research examining circadian clocks in those with Alopecia areata? If so, this would be an interesting point for discussion. If not, this should be added as an area of research lacking in the field. The authors briefly mention androgenetic alopecia, but what about alopecia related to the immune system. The immune system is intertwined with the circadian clock as well.
- Has there been any research with female mice/cells and the effect of the clock on HFs? The authors mentioned some sampling of female human workers, but this seems limited to night shift studies. From a mechanistic point of view, is the clock effects/regulation of HFs the same in males and females? Or is this still unknown. This would be an important point of discussion. Potentially a limitation in this field. Utilization HF to determine chronobiology will be limited if both sexes have not been fully studied.
- Has there been any reported research on feeding cues and rhythms in HFs? Many shift workers eat at irregular times due to their schedules, while some maintain a “normal” feeding schedule despite working the night shift.
- The manuscript should be edited throughout for grammatical errors.
o CLOCK misspelled on line 172
Round 2
Reviewer 2 Report
The authors were very responsive to my previous comments, and I believe the manuscript has improved greatly.